# Quality and outcomes in global cancer surgery: protocol for a multicentre, international, prospective cohort study (GlobalSurg 3)

NIHR Global Health Research Unit on Global Surgery

**Correspondence to**
Ewen Harrison;
ewen.harrison@gmail.com

## ABSTRACT

**Introduction** Empirical, observational data relating to the diagnosis, management and outcome of three common worldwide cancers requiring surgery is lacking. However, it has been demonstrated that patients in low/middle-income countries undergoing surgery for cancer are at increased risk of death and major complications postoperatively. This study aims to determine quality and outcomes in breast, gastric and colorectal cancer surgery across worldwide hospital settings.

**Methods and analysis** This multicentre, international prospective cohort study will be undertaken by any hospital providing emergency or elective surgical services for breast, gastric or colorectal cancer. Centres will collect observational data on consecutive patients undergoing primary emergency or elective surgery for breast, gastric or colorectal cancer during a 6-month period. The primary outcome is the incidence of mortality and major complication rate at 30 days after cancer surgery. Infrastructure and care processes in the treatment of these cancers worldwide will also be characterised.

**Ethics and dissemination** This project will not affect clinical practice and has been classified as clinical audit following research ethics review. The protocol will be disseminated through the international GlobalSurg network.

**Trial registration number** NCT03471494; Pre-results.

## Strengths and limitations of this study

► This will be the first international, multicentre, prospective study to assess quality and outcomes in patients undergoing surgery for three of the most common global cancers.

► The collaborative methodology adopted by our group, as described elsewhere, has previously delivered two large high-quality studies, while avoiding overburdening low-resource centres that may otherwise be unable to participate in such projects.

► Definitions of quality in surgical cancer care are disputed and little evidence exists of their validity or appropriateness in low- and middle-income countries; high-quality data will help identify specific measures for cancer care in resource-limited settings.

► Only those patients undergoing primary surgery for breast, gastric or colorectal cancers will be included, and therefore outcomes in patients receiving only conservative or oncological therapy will not be included.

► As strict primary data monitoring is not possible within the limitations of the study, we will use a previously developed mixed-methods validation process.

## INTRODUCTION

Of the 15.2 million individuals diagnosed with cancer in 2015, 80% required surgery.[1] In tumours amenable to surgical resection, surgery often offers the best chance of cure, particularly in early stage disease. It has been estimated that 45 million surgical procedures are needed each year worldwide, yet fewer than 25% of patients with cancer have access to safe, affordable and timely surgery. While death rates from cancer are decreasing in high-income countries, the opposite has been demonstrated in low/middle-income countries (LMICs).[2] Up to 1.5% of the gross domestic product is lost because of cancer in some LMIC regions.[3]

Our recent LMIC-led three-stage research prioritisation exercise identified cancer surgery as a major research priority. Breast cancer, gastric cancer and colorectal cancer represent a significant burden of disease across income settings.[1 2] Yet, most studies that examine the global distribution and outcomes of solid cancers use simulated methods due to the absence of robust data, including country-specific information on cancer epidemiology, stage distribution and treatment approaches.[1]

Our previous prospective, observational cohort studies GlobalSurg 1 and 2[4 5] have demonstrated that patients in LMICs have an increased risk of death and complications following gastrointestinal cancer surgery. These differences persisted in multivariable

models accounting for confounders in mortality (OR 3.18, 95% CI 2.12 to 4.76), major complication (OR 2.14, 95% CI 1.19 to 3.84) and SSI (OR 1.32, 95% CI 1.04 to 1.68) at 30 days after surgery. Postoperative complications can have a more severe consequences in LMICs, including death, long-term disability and catastrophic healthcare expenditure.[6]

The measures used to determine the quality of surgical cancer care are controversial and subject to ongoing debate. Guidelines produced by bodies such as the National Institute for Health and Care Excellence (UK) and American College of Surgeons in high-income countries provide some consensus.[7 8] However, there is little evidence on the appropriateness of such guidelines in LMICs or what specific measures may indicate quality in cancer surgery in resource-poor settings.

The aim of the GlobalSurg 3 study is to determine variation in the quality of cancer surgery worldwide, focusing on patient outcomes, infrastructure and care processes. This study is driven from within our well-established global network and will be performed in upwards of 85 countries.

### Primary aims
The primary aim is to audit 30-day mortality and complication rates after cancer surgery across low-human, middle-human and high-human development index (HDI) countries.

### Secondary aims
The secondary aim is to measure the quality of surgical cancer care and is designed to be relevant in low-income, middle-income and high-income settings. Conditional data points will be dependent on the specific resources available in a hospital and will include infrastructure, care process measures and outcomes.

### METHODS AND ANALYSIS
This is a multicentre, international, prospective, observational cohort study of all consecutive patients undergoing surgery for breast, gastric or colorectal cancer over a 28-day period. Individual collaborators are free to choose any 28-day period within the 6-month study period to collect data. This 'snapshot' study design is a validated model that has been delivered successfully in previous studies.[4 5 9]

### The research collaborative
GlobalSurg (http://globalsurg.org/) is a collaboration between practising surgeons from around the world, performing research in surgery to foster local, national and international research networks. The collaborative model used has previously been described elsewhere[10] and has already facilitated two multicentre, international, prospective cohort studies including a total of 26 228 patients undergoing emergency and elective abdominal surgery.[4 5] The NIHR Unit on Global Surgery was

established in 2017 and is a consortium between the Universities of Birmingham, Edinburgh and Warwick, together with international partners. The units objective is to advance the education of medical students and doctors in surgical science, clinical research and audit methods by promoting participation in collaborative clinical research and audit studies.

### Study setting
Any surgical unit providing emergency or elective surgery for breast, gastric or colorectal cancer worldwide is eligible to participate. An eligible hospital is not required to perform surgery for all three conditions; however, consecutive patients with breast, gastric or colorectal cancer managed surgically in an individual centre must be collected during the specified study period.

Included centres must capture all consecutive patients and ensure data collection is >90% complete. Centres with >10% missing data, when including all data points, will be excluded from the final analysis and removed from the authorship. There is no minimum number of patients per centre, as long as all eligible patients treated during the study period are included. Multiple teams covering different non-overlapping time periods at each hospital are encouraged.

### Patient inclusion and exclusion criteria
Adult patients aged 18 years or over undergoing emergency or elective surgery for breast, gastric or colorectal cancer are eligible to enter. Any operative approach or treatment intent can be used. Patients whose primary pathology is not suspected to be breast, gastric or colorectal cancer; have a recurrence of their cancer; or are undergoing a procedure that does not require a skin incision should be excluded (box 1). Each individual patient should only be included once into the study.

---

> **Box 1  Patient inclusion and exclusion criteria**
>
> **Inclusion criteria**
> ► Adult patients aged 18 years or over.
> ► Consecutive patients undergoing therapeutic surgery (curative or palliative) for breast, gastric and colorectal cancer.
> ► Patients with suspected benign pathology preoperatively whom were subsequently found to have a diagnosis of cancer following their surgery.
> ► Undergoing emergency or elective procedure requiring a skin incision performed under general or neuraxial (eg, regional, epidural or spinal) anaesthesia.
> ► Includes open, laparoscopic, laparoscopic converted and robotic cases.
>
> **Exclusion criteria**
> ► Operations with a sole diagnostic or staging intent.
> ► Procedures which do not require a skin incision.
> ► Patients with recurrence of breast, gastric or colorectal cancer.

---

### Box 2  Clavien-Dindo classification of major postoperative complications[11]

**Clavien-Dindo grade III**
► Unplanned surgical, endoscopic or radiological intervention.
  – IIIa: intervention not under general anaesthesia.
  – IIIb: intervention under general anaesthesia.

**Clavien-Dindo grade IV**
► Life-threatening complication requiring unplanned critical care management.
  – IVa: single organ dysfunction (including dialysis).
  – IVb: multiorgan dysfunction.

### Outcome measures

The primary outcome measure is the rate of mortality and major complication within 30 days of surgery. Major complications will be defined as occurrence of a Clavien-Dindo[11] grade III or IV (box 2) complication within 30 days of index operation, where day of operation is day 0.

The secondary outcomes that will be derived from this study include incidence of surgical site infection and predefined cancer-specific quality measures for infrastructure and outcomes in cancer care (box 3–5).

### Data points

Data points relating to patient characteristics, cancer staging, neoadjuvant therapy, operative treatment and postoperative period will be collected (online supplementary files 1–4). In order to maximise data completion,

### Box 3  Breast cancer quality measures

**Infrastructure and care processes**
*Availability and performance of:*
► Preoperative fine needle aspiration/core biopsy to diagnose breast cancer.
► Breast/axillary MRI for staging.
► Breast conservation surgery for American Joint Committee on Cancer (AJCC) stage 0/I/II breast cancer.
► Axillary/breast radiotherapy and axillary lymph node clearance (at least 10 lymph nodes for analysis).
► Sentinel lymph node biopsy for early invasive breast cancer.
► Progesterone receptor, oestrogen receptor, human epidermal growth factor receptor 2 receptor and Ki67 status for invasive cancers.
► Treatment with adjuvant treatment where appropriate within 31 days of completion of surgery.
► Plan for radiotherapy for all with breast conserving surgery with clear margins (including ductal carcinoma in-situ [DCIS]).
► Treatment decisions made within multidisciplinary team meeting/tumour board.

**Outcomes**
► 30-day complication rate of surgical site infection, abscess formation, seroma, unplanned reoperation, unplanned readmission and requirement for unplanned critical care.
► Margin involvement (or ability to measure this locally) with 'tumour on inked margin' or a margin <2 mm in DCIS considered positive.

### Box 4  Gastric cancer quality measures

**Infrastructure and care processes**
*Availability/performance of:*
► Endoscopy and biopsy to reach a diagnosis of cancer.
► CT chest, abdomen and pelvis scan performed for preoperative staging.
► Preoperative or postoperative chemotherapy for gastric cancer.
► Treatment decisions made within multidisciplinary team meeting/tumour board.

**Outcomes**
► 30-day complication rate of surgical site infection, anastomotic leak, unplanned reoperation and requirement for unplanned critical care.
► At least 15 regional lymph nodes removed and pathologically examined for resected gastric cancer (or ability to measure this locally).

a minimal dataset has been designed including factors only relevant to quality and outcome measures in surgery for cancer. Review by international collaborators within the GlobalSurg Collaborative has also ensured the dataset is relevant to cancer surgery in a worldwide setting. Investigators will enter data via the secure internet-based Research Electronic Data Capture (REDCap) system.[12] Anonymous patient data will be held on the system hosted by the University of Edinburgh, Scotland, UK.

### Investigators

The study will be undertaken by investigators around the world who will be responsible for disseminating the protocol at their individual site, ensuring appropriate study approvals are in place, identifying and including all eligible patients during each 4-week data collection period and responsible for accurate uploading of data to an online REDCap database.

### Box 5  Colorectal cancer quality measures

**Infrastructure and care processes**
*Availability/performance of:*
► CT chest, abdomen and pelvis scan performed for preoperative staging.
► Preoperative MRI for rectal cancer.
► Planning and treatment with postoperative chemotherapy following resection for lymph node positive colon cancer.
► Treatment with preoperative chemotherapy/radiotherapy.
► Treatment decisions made within multidisciplinary team meeting/tumour board.
► Stoma formation rate.

**Outcomes**
► 30-day complication rate of surgical site infection, anastomotic leak, unplanned reoperation, unplanned readmission and requirement for unplanned critical care.
► Circumferential resection margin >1 mm (or ability to measure this locally).
► At least 12 regional lymph nodes removed and pathologically examined for resected colon cancer (or ability to measure this locally).

A central study writing committee comprising of an internationally representative group of healthcare professionals will be responsible for data analysis, final manuscript drafting and submission. Individuals will be required to register their unit via the REDCap system and will be required to complete a training module prior to commencing data collection.

Countries with multiple sites will be assigned a country lead, who will be responsible for coordinating multiple teams across sites to ensure duplication of data does not occur. Where individual hospitals have a large number of local coordinators, a hospital lead will be appointed to aid coordination. A maximum of three local investigators can cover each 4-week data collection period, with the collection of multiple, non-overlapping collection periods by the same or different local investigators in a single centre possible. They will be responsible for gaining local audit, service evaluation or research ethics approval as appropriate to their institution.

Investigators will create clear mechanisms appropriate to their institution to identify and include all eligible patients, involving daily review of operating logbooks, multidisciplinary team meeting, admission and handover lists. This will include identifying clear pathways to accurately collect baseline, cancer-specific and follow-up data within the normal limits of follow-up. Local arrangements may include daily review of the patient and notes focused on included data points, reviewing patient status in outpatient clinics or via telephone interview at 30 days (if this is normal practice) and checking for readmission through handover lists. All investigators will be listed as collaborators on resulting publications in accordance with previous consensus guidelines for collaborative group research.[13]

### Quality of data
To ensure high data quality, a detailed protocol has been produced and published online. Translations into 12 common languages has also been performed to ease investigator understanding, including Arabic, French, Hindi, Italian, Mandarin, Portuguese, Russian, Spanish and Swahili. Collaborators are encouraged to perform data input in real-time using the REDCap system, with an individual patient record requiring to be completed before submission is possible. Data quality rules will also ensure data quality, highlighting disparities in data fields to the local collaborator for review. Online training is available to collaborators prior to the commencement of data collection at their institution, detailing secure REDCap data entry, patient outcome assessment and disease-specific parameters.

### Data validation
Data validation will be performed in two parts across a group of representative centres similar to the structure successfully used in previous studies of this nature.[5] Case ascertainment assessment will involve an independent investigator determining the number of eligible cases within a 4-week data collection centre and comparing this to the actual number of cases submitted. By comparing samples, a quantitative estimate of case ascertainment will be produced by the central data team. Second, validators will be asked to provide data for a subset of variables, two patient variables, two operation variables and two outcome measures in order to measure data accuracy.

### Statistical analysis and power calculation
Variation across different international health settings will be tested using the HDI,[14] a composite statistic of life expectancy, education and income indices published by the United Nations. Bayesian multilevel logistic regression models will be constructed to account for case mix, with population stratification by hospital and country of residence incorporated as random effects with constrained gradients.

Further prespecified subgroup analyses will be made by geographical country grouping, cancer-type (including the separation of colonic and rectal tumours), emergency versus elective surgery, performance status, palliative versus curative surgery, extent of staging and extent of pathological analyses. When assessing quality measures and processes similar patient groups will be compared, with potential confounding factors such as cancer-type, patient presentation, surgical intent and availability of adjuvant therapy accounted for within statistical models. Quality metrics as described earlier in the protocol will guide exploratory analysis into the global variation in surgical management and available resources. However, it is acknowledged that such guidelines, in the majority, are designed for high-income settings and therefore their attainment will not be considered mandatory or a potential definitive measure of care quality in global cancer surgery.

Data will not be analysed or reported at an individual surgeon or hospital level. Following analysis, results will be fed back to participants at the centre level, but no other centres will be identifiable.

Estimates of 30-day mortality for gastrointestinal cancer resection were determined using data from the GlobalSurg 1 and 2 studies.[4 5] Stratification of results by HDI was performed, with prominent variation in 30-day mortality rate between high HDI and low/middle HDI groups seen after cancer surgery in both emergency surgery (11.6% [75/644] vs 27.3% [59/216]) and elective surgery (2.0% [30/1501] vs 5.5% [23/416]). An indicative sample size calculation using the smaller of these estimates suggests around 500 patients per group at 80% power (P1=0.020, P2=0.055, alpha=0.05) or 640 patients per group at 90% power would be required to conclude a difference in 30-day mortality rate between HDI groups.

### Patient and public involvement
Patient representatives for GlobalSurg, from both the UK and Rwanda, guided development of the research question, outcomes measured and study design. Patients were not involved in the recruitment or conduct of the study.

**Box 6    Study audit standards**

**Breast cancer**
► American College of Surgeons Commission of Cancer Quality of Care for Breast Cancer.[8]
► National Institute for Health and Care Excellence (NICE): early and locally advanced breast cancer: diagnosis and treatment; Clinical Guideline CG80.[17]
► The Society of Surgical Oncology and the American Society for Radiation Oncology (SSO-ASTRO) consensus guidelines for early stage breast cancer.[18]

**Gastric cancer**
► American College of Surgeons Commission of Cancer Quality of Care for Gastric Cancer.[8]
► NICE: oesophago-gastric cancer: assessment and management in adults.[19]

**Colorectal cancer**
► American College of Surgeons Commission of Cancer Quality of Care for Colorectal Cancer.[8]
► NICE: colorectal cancer: diagnosis and management; Clinical Guideline CG131.[20]

We aim to publish the study results as open access, which will be readily available to patients and the public.

## ETHICS AND DISSEMINATION
### Research ethics approval
The primary audit standards stems from the UK National Institute for Health Clinical Excellence[7] and the American College of Surgeons Commission of Cancer Quality of Care[8] guidelines for the diagnosis, investigation and management of breast, gastric and colorectal cancer (box 6). As this study will not change local clinical practice and is limited to using data obtained as part of usual care, it has been classified as an audit by the South Scotland Research Ethics Service in Edinburgh, Scotland (online supplementary file 5). Therefore, this may be considered a global audit or global service evaluation. Local investigators will be responsible for ensuring the study is registered appropriately and approval gained from the relevant local clinical audit departments, research and development department or institutional review boards. If such departments are unavailable, written permission should be supplied by the chief of surgery or responsible supervising consultant/attending physician.

### Protocol dissemination
The protocol will be disseminated across the established GlobalSurg network, compromised of surgeons, medical students and clinical staff across the world. The network previously included over 1800 collaborators across 343 centres representing 66 countries.[5] Country leads are responsible for local coordination and dissemination within their country. In addition, the use of social media including Facebook, Twitter and YouTube has been shown to be an effective medium for dissemination of such collaborative projects[15] and will also be employed.

### Dissemination of results
We aim to publish the study results as open access. Data from the study will be described to ensure individual countries, hospitals and surgeons are anonymous and then shall be deposited in an online data repository for others to analyse. On completion of the study, participating centres will be provided with their own benchmark performance and access to interactive web-based applications to use for quality improvement or subsequent reaudit. Based on the results of the GlobalSurg 3 study, feasibility studies investigating the collection of other outcome measures relating to cancer surgery and development of quality improvement and/or interventional clinical trials will be suggested for possible implementation in surgical cancer units for each included hospital in the study.

## DISCUSSION
In this study protocol, we describe a multicentre, international, prospective cohort study investigating the quality and outcomes of surgery for three of the most common global cancers. Despite the likely increased risk of mortality and major morbidity for patients undergoing surgery for cancer in LMICs, high-quality, empirical data are currently unavailable. Furthermore, in countries with limited resources applicability of cancer surgery guidelines are yet to be tested.

By using a collaborative methodology and a short 4-week data collection period, the study will recruit sufficient patients to measure this, while avoiding burdening low-resource centres that may otherwise be unable to participate. Investigating the morbidity and mortality caused by cancer surgery globally, this study will provide a platform to build future quality improvement programmes and interventional trials as previously demonstrated by the GlobalSurg network.

This study will be delivered using an international multidisciplinary collaborative network of healthcare researchers, with the collaborative model having consistently proven its ability to produce high-quality outcomes in international studies.[4 5] A detailed study protocol in multiple languages, mandatory training, data quality control and validation period will ensure standardisation to deliver a reliable and accurate data set.

As the second most common cause of death in 2015, with 8.7 million deaths globally,[2] cancer incidence is predicted to become an increasing burden worldwide[1 2] and place further pressure on already limited healthcare systems. Neoplasms already contribute to significant global morbidity and mortality, causing the highest loss of gross domestic product of any surgical disease.[3] Surgery can provide cure for many cancers, particularly in countries where limited access to oncology treatment exists. However, the majority of the world's population lack access to safe, affordable and timely cancer surgery.[16]

This study provides the first opportunity to collect and analyse prospective, observational data for three of the most common global cancers. Current literature is heavily reliant

on simulated models based on limited data sources.[2 3 16] Our study will quantify any global inequalities in cancer surgery, highlight differences in patient presentation, treatment interventions and surgical outcomes.

With feedback of outcomes and specific quality measures relating to each cancer, collaborators will have the opportunity to appraise their current practice against a global standard. Furthermore, surgeons and other interested parties will be able to use the findings from this study to help develop focused cancer surgery guidelines based on empirical global data.

Finally, this study will continue to strengthen the international GlobalSurg network, further developing capacity for research in LMICs. Focused interventional trials derived from study findings will follow, aimed at improving global outcomes in cancer surgery.

**Acknowledgements**  We would like to thank Emmy Runigamugabo and Azmina Verjee for their advice and contribution as patient representatives in the development of this study protocol.

**Collaborators**  Stephen R Knight; Thomas M Drake; Dmitri Nepogodiev; J Edward F Fitzgerald; Adesoji O Ademuyiwa; Philip Alexander; Jean C Allen Ingabire; Sara W. Al-Saqqa; Bruce Biccard; Giuliano Borda; David Borowski; Sule Burger; Kathryn Chu; Damian Clarke; Ainhoa Costa; Justine Davies; Rachel Donaldson; Chikwendu Ede; O James Garden; Dhruv Ghosh; James C. Glasbey; T Peter Kingham; Hosni Khairy Salem; Anyomih Theophilus Kojo; Zach Koto; Marie Carmela Lapitan; Ismail Lawani; Chiapo Lesetedi; Maria Lorena Aguilera; Charles Mabedi; Mayaba Maimbo; Laura Magill; Felix Makinde Alakaloko; Alex Makupe; Janet Martin; Antonio Ramos-De la Medina; Mark Monahan; Rachel Moore; Vanessa Msosa; Soloman Mulira; Alphonse Zeta Mutabazi; Elmi Muller; Joseph Musowoyo; Adewale Oluseye Adisa; Jean Léon Olory-Togbe; Riinu Ots; Ahmad Uzair Qureshi; Sarah Rayne; Tracey Roberts; Marie Dione Parreno-Sacdalan; Catherine Shaw; Neil Smart; Martin Smith; Richard Spence; Steph van Straten; Stephen Tabiri; Viki Tayler; Thomas G Weiser; John Windsor; Joseph Yorke; Raul Yepez; Richard Lilford; Dion Morton; Aneel Bhangu; Sudha Sundar; Ewen M Harrison.

**Contributors**  All authors within the NIHRGHRUGS contributed to the design, drafting and review of this study protocol. In addition, EH is also the overall guarantor for the article.

**Funding**  This study was funded by Department for International Development (DFID)—Medical Research Council (MRC)—Wellcome Trust Joint Global Health Trial Development Grant (MR/N022114/1). A National Institute of Health Research (NIHR) Global Health Research Unit Grant (NIHR 17-0799) is supporting the establishment of surgical research units in a subset of contributing low-income countries.

**Disclaimer**  The views expressed are those of the authors and not necessarily those of the NHS, the NIHR or the UK Department of Health and Social Care.

**Competing interests**  GlobalSurg is run by the Surgical Research Gateway (SuRG) Foundation. The SuRG Foundation is a registered charity (charity number 1159898) whose object is to advanced the education of medical students and doctors in surgical science, clinical research and audit methods by promoting participation in collaborative clinical research and audit studies.

**Patient consent for publication**  Not required.

**Ethics approval**  South East Scotland Research Ethics Service in Edinburgh.

**Provenance and peer review**  Not commissioned; externally peer reviewed.

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
