## [Reviewer comments · BMJ Open]

ARTICLE DETAILS

TITLE (PROVISIONAL)	Quality and outcomes in global cancer surgery: protocol for a multicentre, international, prospective cohort study (GlobalSurg 3)
AUTHORS	on Global Surgery, NIHR Global Health Research Unit; Harrison, Ewen

VERSION 1 - REVIEW

REVIEWER	Alex Haynes, MD, MPH Ariadne Labs, Massachusetts General Hospital, Harvard Medical School Boston, USA
REVIEW RETURNED	17-Oct-2018

GENERAL COMMENTS	This is an exciting third iteration of the GlobalSurg team, looking at important issues of surgery on an international basis. I am glad to see that the collaborative approach is being applied to cancer surgery to better understand the heterogeneity of care provided throughout the world and look forward to the results of the trial. The overall approach mirrors that of the previous GlobalSurg studies. I do have a number of questions about the numerous quality metrics that have been selected for the three cancer types, as well as some questions about inclusion criteria. 1. Presumably, elective and urgent/emergent cases will be assessed separately when looking at process measures. It would not be appropriate, for example, for a formal multidisciplinary tumor board to be consulted for a patient presenting with a perforated colon cancer. Similarly, different conclusions might be made in terms of needed interventions for two sites with similar surgical mortality rates but very different rates of urgent presentation.2. Similarly, patients operated on for palliative intent might be very different. A patient who presents with obstructing rectal cancer and widespread metastases would be expected to have a different procedure performed and different processes followed than one who presents electively for resection of a stage II tumor. Particularly for assessment of care processes, I would urge that you consider the primary study group those undergoing curative intent surgery in an elective setting. Very reasonable to collect and look at others, but it would be misleading to evaluate all together, especially across such heterogeneous sites as included in this study.3. I am interested in how the quality metrics for each disease site were selected and it isn't clear how precisely they are assessed. For example, are the authors suggesting that all patients with
--

	stage 0/II breast cancer be treated with breast conserving therapy? It is perfectly appropriate in most cases to treat with mastectomy and there are some instances where it would be inappropriate to treat with BCS (e.g. male patients, large tumors in a small breast, tumors very near the skin or chest wall, patients with prior radiation). There are numerous other quality metrics that may not truly be markers of high quality care in all instances: breast MRI, axillary/breast radiotherapy and axillary lymph node clearance (for whom? all patients?), timing of adjuvant therapy). Perhaps I am missing something in how these are defined. I can see things like "not indicated" on the data sheets, but it is not clear how that would be adjudicated. 4. There are similar questions about the quality metrics for gastric and colorectal cancer, particularly around neoadjuvant and adjuvant therapy indications. There is some controversy about who should receive these under which circumstances and I would be reluctant to make a strong statement about this in the context of this study. 5. While colon and rectal cancer are often grouped together, they really represent different diseases in terms of their treatment and prognosis. I would urge you to separate them out for all analyses and separate the two sets of quality indicators as this will simplify things for analysis. I hope that these comments are helpful and that this study can help illuminate the state of cancer surgery on a global basis.
--	---

REVIEWER	Professor David Watters Deakin University and Barwon Health Victoria, Australia
REVIEW RETURNED	07-Nov-2018

GENERAL COMMENTS	Global cancer burden and treatment are a major issue to address during the period of the SDG's to 2030 as reference 1 suggests. This study protocol is of value, not only because it describes a methodology that can be repeated in these three cancers chosen (breast, gastric and colorectal), but also because the methodology could be applied to other cancers in subsequent studies. This Global Surgery collaborative already has a track record for publication and multiple data entry from various countries and collaborators. They have been successful with previous research questions and they are likely to succeed with this one. The results will be interesting and informative when published. However, the methodology is just as valuable which is why I support this paper being published and made widely available. Finally, I support the way that the protocol team involve multiple contributors and authors from different countries, and facilitate young researchers willing to collect data being engaged in clinical research. Research capability is still poor in many developing countries and this research is likely to enhance and strengthen local research capability, and challenge countries to review their own data and discuss what it means.
---

VERSION 1 – AUTHOR RESPONSE

Reviewer 1

This is an exciting third iteration of the GlobalSurg team, looking at important issues of surgery on an international basis. I am glad to see that the collaborative approach is being applied to cancer surgery to better understand the heterogeneity of care provided throughout the world and look forward to the results of the trial. The overall approach mirrors that of the previous GlobalSurg studies. I do have a number of questions about the numerous quality metrics that have been selected for the three cancer types, as well as some questions about inclusion criteria.

1. Presumably, elective and urgent/emergent cases will be assessed separately when looking at process measures. It would not be appropriate, for example, for a formal multidisciplinary tumor board to be consulted for a patient presenting with a perforated colon cancer. Similarly, different conclusions might be made in terms of needed interventions for two sites with similar surgical mortality rates but very different rates of urgent presentation.

Thank you for highlighting this point. We will distinguish between elective and emergency cases when performing analyses to provide an accurate and unbiased interpretation of measured outcomes and process measures. We have further clarified this point in the statistical analysis and power calculation section.

2. Similarly, patients operated on for palliative intent might be very different. A patient who presents with obstructing rectal cancer and widespread metastases would be expected to have a different procedure performed and different processes followed than one who presents electively for resection of a stage II tumor. Particularly for assessment of care processes, I would urge that you consider the primary study group those undergoing curative intent surgery in an elective setting. Very reasonable to collect and look at others, but it would be misleading to evaluate all together, especially across such heterogeneous sites as included in this study.

We plan to account for these variables within our analysis and agree it is important to compare homogenous patient groups in order to avoid confounding and strengthen conclusions. There is an issue around patients undergoing a planned curative resection who are subsequently labelled “palliative” when things don’t go well. This has been well-described in high income settings as a method for manipulating outcome data. However, the point is well made. We have inserted additional text around this point in the Statistical analysis and power calculation section, (paragraph 2).

3. I am interested in how the quality metrics for each disease site were selected and it isn't clear how precisely they are assessed. For example, are the authors suggesting that all patients with stage 0/I/II breast cancer be treated with breast conserving therapy? It is perfectly appropriate in most cases to treat with mastectomy and there are some instances where it would be inappropriate to treat with BCS (e.g. male patients, large tumors in a small breast, tumors very near the skin or chest wall, patients with prior radiation). There are numerous other quality metrics that may not truly be markers of high quality care in all instances: breast MRI, axillary/breast radiotherapy and axillary lymph node clearance (for whom? all patients?), timing of adjuvant therapy). Perhaps I am missing something in how these are defined. I can see things like "not indicated" on the data sheets, but it is not clear how that would be adjudicated.

The measures of quality relating to cancer surgery have been selected from the National Institute for Health and Care Excellence (NICE) and American College of Surgeons, however we acknowledge that these guidelines are based in high-income settings and concur that these may not be applicable or implementable within LMIC settings. As the reviewer emphasises, individual treatment decisions can deviate from relevant guidelines for valid clinical reasons.

The main aim of the study is to examine 30-day outcomes (death and major morbidity) following cancer surgery, but we also hope to examine and demonstrate the global variability in surgical treatment and available resources to patients with cancer amenable to surgical management. The selected guidelines will allow for the global comparison of treatment for three common cancers, however, in depth conclusions may be limited for the reasons raised.

Furthermore, we agree that the presence of CT, MRI, tumour markers and selection of particular surgical interventions may not equate to high quality cancer care in particular circumstances, but currently data exploring these metrics is limited. We hope to explore these points within our analysis, with the potential to develop more specific quality parameters based on the availability and impact of individual variables on patient outcomes.

We have aimed to pragmatically capture data on pre-operative imaging, staging investigations and oncology treatment variables by including numerous options for local collaborators to select for each individual case. These responses include the following:

- Patient does not need it
- No patient needs it, but not available
- No, patient needs it, facilities available, but patient not able to pay
- No, planned but not given
- Unknown

These will be recorded by local collaborators according to local treatment practices and case presentation, which will ultimately be influenced by factors such as affordability, availability and patient characteristics. We acknowledge that these responses may be open to interpretation by collaborators, however consensus between collaborators (within their mini-teams) is required prior to data entry completion and will act to limit variability in responses.

We have added further content within the manuscript and hope this has addressed the reviewers concerns.

4. There are similar questions about the quality metrics for gastric and colorectal cancer, particularly around neoadjuvant and adjuvant therapy indications. There is some controversy about who should receive these under which circumstances and I would be reluctant to make a strong statement about this in the context of this study.

We agree this is important. There will need to be careful interpretation of the data. We aim to present this objectively and may pass some of the value judgement to report readers. For instance, if a young fit patient with a node positive colon cancer does not receive adjuvant chemotherapy, readers will make a judgement about whether that care should be considered high quality or not.

As the reviewer highlights debate exists around oncological treatment in these cancers and strong statements will be avoided, particularly as this is a secondary aim of the study.

5. While colon and rectal cancer are often grouped together, they really represent different diseases in terms of their treatment and prognosis. I would urge you to separate them out for all analyses and separate the two sets of quality indicators as this will simplify things for analysis.

Site of colorectal cancer will be included as a variable in multivariable analyses. We have added a statement to this effect within the manuscript (Statistical analysis and power calculation section, paragraph 2).

Reviewer 2

Global cancer burden and treatment are a major issue to address during the period of the SDG's to 2030 as reference 1 suggests. This study protocol is of value, not only because it describes a methodology that can be repeated in these three cancers chosen (breast, gastric and colorectal), but also because the methodology could be applied to other cancers in subsequent studies.

This Global Surgery collaborative already has a track record for publication and multiple data entry from various countries and collaborators. They have been successful with previous research questions and they are likely to succeed with this one.

The results will be interesting and informative when published. However, the methodology is just as valuable which is why I support this paper being published and made widely available.

Finally, I support the way that the protocol team involve multiple contributors and authors from different countries, and facilitate young researchers willing to collect data being engaged in clinical research. Research capability is still poor in many developing countries and this research is likely to enhance and strengthen local research capability, and challenge countries to review their own data and discuss what it means.

Thank you for your comments and positive review. We look forward to reporting the results from GlobalSurg 3 and, as you highlight, hope our methodology will allow for the further exploration of global outcomes following surgery in other cancers.

VERSION 2 – REVIEW

REVIEWER	Alex Haynes Massachusetts General Hospital and Ariadne Labs, Harvard Medical School
REVIEW RETURNED	08-Jan-2019

GENERAL COMMENTS	Revisions have added to the clarity. The quality of care metrics will remain sticky to parse out and I still remain concerned about some potentially problematic emphasis on NICE metrics (developed for use in the UK) that may not be applicable in other settings. This could even have untoward outcomes (e.g. an emphasis on breast conserving surgery for breast cancer in settings where access to radiotherapy is limited may lead to either omission of adjuvant RT (an essential component of breast conserving therapy) or use of substandard RT with additional patient risk. With that being said, this is a worthy undertaking and I trust that the authors will be circumspect in how they present and interpret the quality data.
---